# Associations between women's empowerment and child development, growth, and nurturing care practices in sub-Saharan Africa: A cross-sectional analysis of demographic and health survey data

Lilia Bliznashka[1]*, Ifeyinwa E. Udo[2], Christopher R. Sudfeld[1,3], Wafaie W. Fawzi[1,3,4], Aisha K. Yousafzai[1]

1 Department of Global Health and Population, Harvard T.H. Chan School of Public Health, Boston, Massachusetts, United States of America, 2 Center for Interdisciplinary Research on AIDS, Yale University School of Public Health, New Haven, Connecticut, United States of America, 3 Department of Nutrition, Harvard T.H. Chan School of Public Health, Boston, Massachusetts, United States of America, 4 Department of Epidemiology, Harvard T.H. Chan School of Public Health, Boston, Massachusetts, United States of America

* lilia.bliznashka@gmail.com

## Abstract

### Background

Approximately 40% of children 3 to 4 years of age in low- and middle-income countries have suboptimal development and growth. Women's empowerment may help provide inputs of nurturing care for early development and growth by building caregiver capacity and family support. We examined the associations between women's empowerment and child development, growth, early learning, and nutrition in sub-Saharan Africa (SSA).

### Methods and findings

We pooled data on married women (15 to 49 years) and their children (36 to 59 months) from Demographic and Health Surveys that collected data on child development (2011 to 2018) in 9 SSA countries ($N = 21,434$): Benin, Burundi, Cameroon, Chad, Congo, Rwanda, Senegal, Togo, and Uganda. We constructed a women's empowerment score using factor analysis and assigned women to country-specific quintile categories. The child outcomes included cognitive, socioemotional, literacy–numeracy, and physical development (Early Childhood Development Index), linear growth (height-for-age Z-score (HAZ) and stunting (HAZ <−2). Early learning outcomes were number of parental stimulation activities (range 0 to 6) and learning resources (range 0 to 4). The nutrition outcome was child dietary diversity score (DDS, range 0 to 7). We assessed the relationship between women's empowerment and child development, growth, early learning, and nutrition using multivariate generalized linear models.

On average, households in our sample were large (8.5 ± 5.7 members) and primarily living in rural areas (71%). Women were 31 ± 6.6 years on average, 54% had no education, and

**Data Availability Statement:** The data underlying the results presented in the study are publicly available from the DHS Program (http://www.dhsprogram.com). Registration is required to access the data.

**Funding:** The authors received no specific funding for this work.

**Competing interests:** The authors have declared that no competing interests exist.

**Abbreviations:** CFA, confirmatory factor analysis; CI, confidence interval; DDS, dietary diversity score; DHS, Demographic and Health Surveys; ECDI, Early Childhood Development Index; EFA, exploratory factor analysis; GII, Gender Inequality Index; HAZ, height-for-age Z-score; LMICs, low- and middle-income countries; MD, mean difference; MDD, minimum dietary diversity; MICS, Multiple Indicator Cluster Surveys; RR, relative risk; SSA, sub-Saharan Africa; UNICEF, United Nations Children's Fund; WHO, World Health Organization.

31% had completed primary education. Children were 47 ± 7 months old and 49% were female. About 23% of children had suboptimal cognitive development, 31% had suboptimal socioemotional development, and 90% had suboptimal literacy–numeracy development. Only 9% of children had suboptimal physical development, but 35% were stunted. Approximately 14% of mothers and 3% of fathers provided $\geq$4 stimulation activities. Relative to the lowest quintile category, children of women in the highest empowerment quintile category were less likely to have suboptimal cognitive development (relative risk (RR) 0.89; 95% confidence interval (CI) 0.80, 0.99), had higher HAZ (mean difference (MD) 0.09; 95% CI 0.02, 0.16), lower risk of stunting (RR 0.93; 95% CI 0.87, 1.00), higher DDS (MD 0.17; 95% CI 0.06, 0.29), had 0.07 (95% CI 0.01, 0.13) additional learning resources, and received 0.16 (95% CI 0.06, 0.25) additional stimulation activities from their mothers and 0.23 (95% CI 0.17 to 0.29) additional activities from their fathers. We found no evidence that women's empowerment was associated with socioemotional, literacy–numeracy, or physical development. Study limitations include the possibility of reverse causality and suboptimal assessments of the outcomes and exposure.

## Conclusions

Women's empowerment was positively associated with early child cognitive development, child growth, early learning, and nutrition outcomes in SSA. Efforts to improve child development and growth should consider women's empowerment as a potential strategy.

## Author summary

### Why was this study done?

- Nearly 40% of children 3 to 4 years of age in low- and middle-income countries have suboptimal development and growth. Children require multiple health, nutrition, early learning, and care inputs to reach their full developmental and growth potential.

- Women's empowerment may help provide these inputs by increasing shared caregiving, building caregiver capacity, and improving family support. Even though women's empowerment is predictive of better child health, nutrition, and growth, less is known about the relationship between women's empowerment and child development and early learning outcomes.

### What did the researchers do and find?

- We pooled Demographic and Health Surveys data from married women (15 to 49 years) and their children (36 to 59 months) in 9 sub-Saharan African countries to investigate the association between women's empowerment and child development, growth, early learning, and nutrition outcomes.

- Our results showed that higher women's empowerment was predictive of better child cognitive development, growth, and nutrition. More empowered women and their partners also had access to more learning resources and provided more stimulation activities to their child.

**What do these findings mean?**

- Our findings indicate that improving women's empowerment is one potential strategy to help provide the nutrition and early learning inputs children require to reach their full developmental and growth potential.

- Efforts to improve early child development and growth should consider the role of women's empowerment in improving early learning and nutrition outcomes.

- Future research should assess the longitudinal and causal relationships between women's empowerment and child development, growth, early learning, and nutrition outcomes.

## Introduction

About 30% of children less than 5 years of age in sub-Saharan Africa (SSA) are stunted [1], and 40% of children 3 to 4 years of age are not developmentally on track [2,3]. These childhood adversities undermine educational attainment, earnings, and health outcomes later in life, leading to long-term loss of human capital [4,5]. Consequently, investing in early childhood development can improve adult educational, labor market, and health behavior outcomes [6] and thus help reduce long-term social inequalities [7]. Likewise, improving nutrition early in life can reduce stunting and bolster adult intelligence, wages, and schooling outcomes [8].

To reach their full developmental and growth potential, children require a "comprehensive, multisectoral system of services and opportunities" [9]. The Children Surviving and Thriving Framework outlines such a system of proximal and distal factors in the enabling environment for the provision of nurturing care. The essential proximal components combine child health and nutrition, included in the United Nations Children's Fund (UNICEF) conceptual framework of malnutrition as necessary for child survival, and responsive care, early learning opportunities, and security and safety, the additional components of nurturing care required for thriving. Underlying these proximal components is a complex network of family and community enabling environments, and distal social, political, and economic factors [9].

In low- and middle-income countries (LMICs), multigenerational nurturing care interventions, which combine health, nutrition, and nurturing care inputs for children with inputs to empower, support, and enable caregivers to provide nurturing care, are recognized as essential for optimal child development and growth in early life [10]. However, where a large body of literature from LMICs has examined the associations between caregiver empowerment and child survival, health, nutrition, and growth [11–19], associations with other proximal components of the Children Surviving and Thriving Framework remain understudied. For instance, to our knowledge, no evidence exists on whether women's empowerment is associated with responsive care, early learning opportunities, and security and safety. Likewise, only one study has examined the association between women's empowerment and child development in SSA [20]. This study showed that women's empowerment was positively associated with child literacy–numeracy development, but not with child cognitive, socioemotional, or physical development [20]. Furthermore, best practices should be established for designing multigenerational interventions to support caregiver empowerment and promote gender equity since most curricula typically focus solely on women. Only a handful of interventions have aimed to improve men's parenting skills, leaving the role of men in promoting a nurturing care environment

understudied [10]. It is unclear how best to engage men through multigenerational interventions to help change traditional gender roles, support women's empowerment, and promote gender equity.

Given this limited evidence, in this paper, we sought to understand how women's empowerment is associated with child development and growth and their underlying proximal components, specifically early learning and nutrition. We first establish a conceptual framework linking women's empowerment to child development and growth, and then use nationally representative data from Demographic and Health Surveys (DHS) from 9 SSA countries to examine the cross-sectional associations. In addition, we sought to inform the design of multigenerational nurturing care interventions seeking to promote women's empowerment and gender equality.

## Methods

### Conceptual framework

We adopted Kabeer's framework, which defines women's empowerment as "the processes by which those who have been denied the ability to make choices acquire such an ability" [21]. A summary of alternative definitions of women's empowerment stemming from Kabeer's framework is provided in **S1 Text**. Kabeer's framework distinguishes 3 interrelated dimensions of empowerment: resources (preconditions), agency (processes), and achievements (outcomes) [21]. However, not all dimensions are relevant to child development and growth, and including those that are not can lead to null or contradictory results [11–13].

Therefore, we developed a conceptual diagram linking women's empowerment and child development and growth (**Fig 1**). While prior studies assessing the relationship between women's empowerment and child outcomes were grounded in the UNICEF conceptual framework

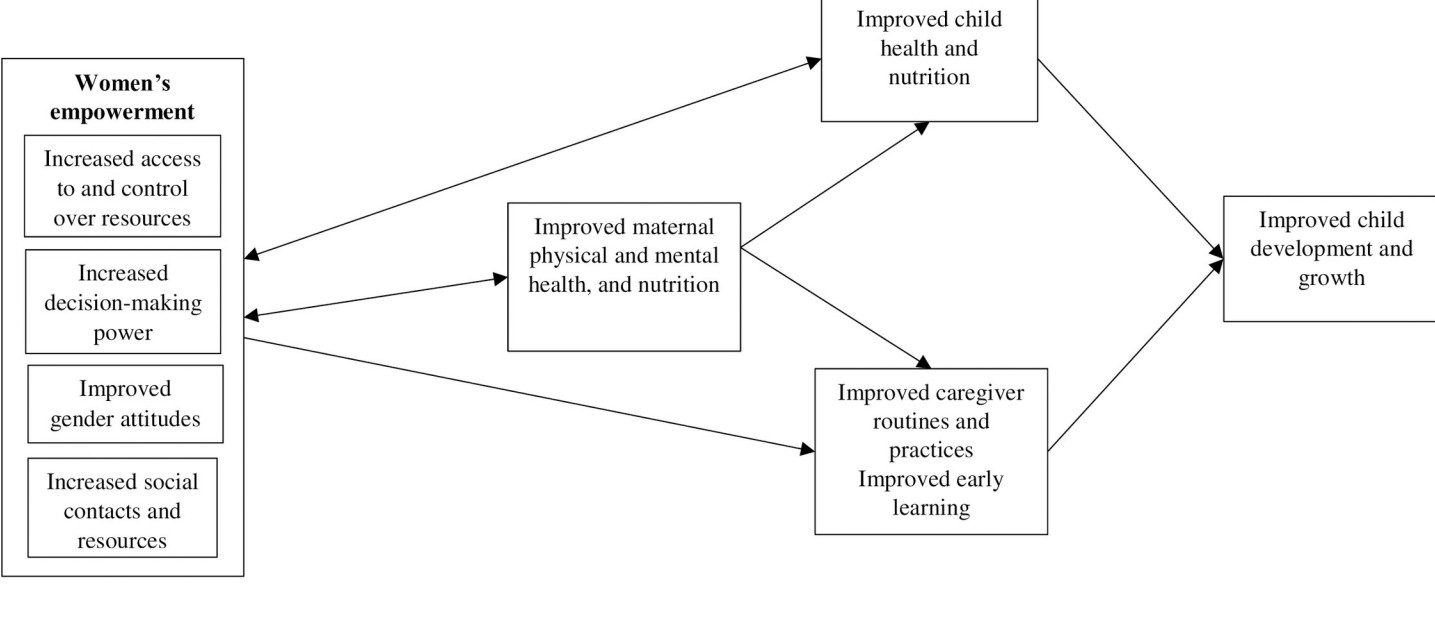

| Pathways | Mediators | Outcomes |

**Fig 1. Pathways linking women's empowerment and early childhood development and growth.** One-directional arrows represent 1-way relationships between women's empowerment, mediators, and outcomes. Bidirectional arrows represent 2-way or interactive relationships between women's empowerment and women's and children's outcomes.

of malnutrition [11,14], we drew on the more recent Children Surviving and Thriving Framework [9] as the theoretical basis for the conceptual diagram. This latter framework includes proximal components of nurturing care (i.e., responsive care, learning opportuning, and security and safety) in addition to the child health and nutrition components serving as the basis of the original UNICEF framework [9]. We delineated the pathways through which women's empowerment can influence these proximal components underlining optimal child development and growth. We extended prior conceptual frameworks beyond child growth to include other proximal components relevant to child development (e.g., early learning).

The conceptual diagram in Fig 1 shows that women's empowerment may influence child development and growth through 4 main pathways: (1) increased access to and control over financial and economic resources; (2) increased social contacts and resources; (3) increased decision-making power; and (4) improved gender attitudes. Women who are employed or involved in income generating activities and women with greater control over assets allocate more resources toward their own and their child's health, nutrition [11,17], and, potentially, development. More empowered women may have more freedom of movement and thus increased access to resources (e.g., by joining savings groups [11]) and ability to obtain resources (e.g., by visiting the health center), as well as ability to perform caregiver routines. Women's employment, and, thus, access to resources, can increase their decision-making capacity in the household [22]. More empowered women may have more say in whether to seek physical and mental health services and whether to use family planning [11]. Through this increased access to health services, more empowered women may learn more about taking care of their own health and translate this knowledge to how they care for their children. More empowered women may also have improved efficiency and effectiveness in making and executing more timely decisions about children's health [14]. They may also be better able to determine the activities (e.g., rest) and resources (e.g., high-quality foods) that are optimal for their own and their children's health and nutrition [11]. More empowered women may have more access to and control over food resources and more decision-making power over intra-household food allocation [11,17]. In addition, more empowered women may have greater say over the type of learning opportunities and materials obtained for their child. Less empowered women may have less say in time allocation and may compromise time allocated to childcare, rest [23], and stimulation activities with their child, whereas more empowered women may have greater say in their own time use. Further, greater empowerment increases the number and frequency of social contacts (either directly through labor force participation or indirectly by increased mobility), which exposes women to new information, behaviors, and attitudes [14,22] and provides social support [23], both of which may improve their care, health, and nutrition practices. Finally, less empowered women may be more socialized to accept inferior gender roles [14] and more likely to experience physical and emotional violence at home or in the community [14,17]. Conversely, more empowered women may be less socialized to accept inferior gender roles, though empirical evidence supporting this hypothesis in LMICs is lacking. Of note is that these pathways are interrelated, and the temporal order in which they influence each other is unclear. In addition, the causal direction between women's empowerment and women's and children's health and nutrition outcomes in our cross-sectional model is ambiguous. For example, women's empowerment may help improve child health outcomes or improved child health may empower women. We express such interactive relationships via bidirectional arrows in Fig 1.

These 4 pathways reflect 4 distinct dimensions of women's empowerment: (1) access to and control over resources, "Resources" for brevity; (2) decision-making; (3) social resources; and (4) gender attitudes, "Gender" for brevity. The "Resources" dimension encompasses women's ownership of and ability to access financial and economic resources (including mobility) and

their ability to allocate such resources toward health, nutrition, growth, development, and early learning opportunities. The "Decision-making" dimension encompasses women's decision-making capacity, power, and timeliness, including decision-making with respect to reproductive health and time use. The "Social resources" dimension represents women's community and social contacts and networks. Finally, the "Gender" dimension encompasses women's intrinsic agency, i.e., voice and ability to express beliefs, and the extent to which women's gender attitudes reflect normative gender beliefs including attitudes toward violence and beliefs around sexual activity [24]. Thus, the "Resources" and "Social resources" dimensions reflect resources, whereas the "Decision-making" and "Gender" dimensions reflect agency.

Together these pathways can help explain the positive associations observed in LMICs between women's empowerment and their own nutritional status [14,16,25–29], diet [25–28,30], mental [31] and physical health; and women's empowerment and children's nutrition [13,14,17,32], health, growth [11–16,33,34], and development [20]. With respect to caregiver routines and practices, and early learning opportunities, evidence is lacking, but links with women's empowerment are plausible based on the pathways just described.

## Data access and ethical considerations

We used deidentified secondary data, which were exempt from full review by the Institutional Review Board of the Harvard T.H. Chan School of Public Health (protocol number IRB20-0402).

## Data and study population

We pooled data from the latest DHS for the following 9 SSA countries, which collected data on child development and were publicly available as of February 21, 2020: Benin (Phase VII, 2017 to 2018, $N$ = 4,305), Burundi (Phase VII, 2016 to 2017, $N$ = 2,123), Cameroon (Phase VI, 2011, $N$ = 1,666), Chad (Phase VII, 2014 to 2015, $N$ = 4,198), Congo (Phase VI, 2011 to 2012, $N$ = 1,342), Rwanda (Phase VII, 2014 to 2015, $N$ = 1,125), Senegal (Phase VII, 2017, $N$ = 4,059), Togo (Phase VI, 2013 to 2014, $N$ = 1,153), and Uganda (Phase VII, 2016, $N$ = 1,463). DHS empowerment questions were designed to measure women's participation in household decision-making, attitudes toward gender equity in roles and rights, and economic activity. Although all women are eligible for the empowerment questions, questions on decision-making are only asked of women who are currently married or cohabitating (referred to as married, for brevity) since 2 response options are only possible if women have a partner/husband [35]. Therefore, we restricted our analysis sample to married women of reproductive age (15 to 49 years).

Child development assessment was added to Phase VI of the DHS using the Early Childhood Development Index (ECDI), which is a population-based measure of child development [36]. ECDI questions are optional, and, when administered, they are collected from a subsample of children 36 to 59 months of age using UNICEF Multiple Indicator Cluster Surveys (MICS) procedures [37]. Therefore, we restricted our sample to children 36 to 59 months of age with available ECDI data. Although the MICS also collect data on child development, growth, and early learning, they do not collect data on women's decision-making in the household or on child diet for children 36 to 59 months of age. Therefore, we did not include MICS data.

We did not register a prospective analysis plan. However, all analyses were planned, and the only data-driven changes that occurred were in the derivation of the women's empowerment score, as a result of conducting exploratory factor analysis (EFA). These were expected and are

described in detail below. Changes in response to reviewers included external validation of the empowerment score and the inclusion of all 4 ECDI domains as outcomes. This study is reported according to the Strengthening the Reporting of Observational Studies in Epidemiology (STROBE) guidelines (**S1 STROBE Checklist**).

## Exposure indicators

We operationalize women's empowerment as a multidimensional latent construct with 4 latent dimensions: (1) Resources; (2) Decision-making; (3) Social resources; and (4) Gender. Since the DHS do not collect data on social resources, we proceeded with a 3 dimensional model of empowerment. We used factor analysis to derive factor scores for the latent women's empowerment dimensions. Although factor analysis and item response theory with binary and categorical variables are formally equivalent, factor analysis is preferred in multidimensional frameworks, with a small number of items, and when several groups are compared for measurement invariance [38–40]. Full methodological details are provided in **S1 Appendix.** We used EFA to evaluate the dimensionality of the women's empowerment construct (results presented in **S2 Appendix**); confirmatory factor analysis (CFA) to test the best-fitting model from the EFA (results presented in **S3 Appendix**); and multigroup CFA to test for measurement invariance across countries to address issues of cross-country comparability of women's empowerment (results presented in **S4 Appendix**). Despite our conceptual framework, using EFA prior to CFA allowed us to more fully explore the latent structure and verify that the 3-factor solution had an acceptable fit and that indicator loadings were generally in line with our hypotheses [41]. EFA results helped refine the CFA specification.

The indicators of the final form-invariant measurement model are presented in **Table 1**. All indicators were coded as binary with 1 representing empowerment, and 0 representing lack of empowerment, except for indicators for work seasonality and income relative to partner, which were coded as categorical variables with higher values indicating higher empowerment. "Don't know" responses were coded as missing. All decision-making indicators were coded as 1 if the woman decided alone or jointly with her husband/partner, and 0 otherwise. However, since it is unclear whether joint decision-making represents disguised male decision-making or cooperation [42], we conducted sensitivity analysis recoding decision-making indicators as 1 if the woman decided alone, and 0 otherwise. Since the "Gender" dimension only contained indicators on attitudes toward wife beating, we renamed this dimension to "Attitudes toward wife beating." The final form-invariant measurement model showed acceptable model fit, based on a priori determined acceptability thresholds [43]: Comparative Fit Index of 0.973 (threshold $\geq$ 0.95), Root Mean Square Error of Approximation of 0.043 (threshold $\leq$ 0.08), and Root Mean Square Error of Approximation of 0.055 (threshold $\leq$ 0.08). This model showed that the indicators measured the same factors in each country; however, the indicators related to income were measured with a different degree of precision. Factor scores for each women's empowerment dimension were estimated from the final form-invariant model. We summed the individual dimension factor scores to create a total empowerment score. The distributions of the individual dimension factor scores and the total empowerment score are shown in **Fig A in S5 Appendix.**

External validity was assessed against the United Nations Gender Inequality Index (GII). The GII is a country-level index that measures gender inequity in reproductive health, political empowerment, and economic activity. A higher GII value indicates more gender disparity [44], and we therefore expected a negative correlation between our empowerment measures and the GII. We estimated country-level means for the individual dimension and total empowerment scores, adjusting for the complex survey design using country-specific sampling

**Table 1. Indicators comprising the latent dimensions of the form-invariant women's empowerment measure.**

| Dimension | Indicator | DHS question | Response codes from lower to higher empowerment |
|---|---|---|---|
| **Access to and control over resources** | Seasonality | Do you usually work throughout the year, or do you work seasonally, or only once in a while? | Not working, seasonal or occasional, all year |
| | Income relative to partner | Would you say that the money that you earn is more than what your (husband/partner) earns, less than what he earns, or about the same? | Does not earn cash, husband/partner has no earnings, less than him, about the same, more than him |
| **Decision-making** | Decision on partner's income use | Who usually decides how your (husband's/partner's) earnings will be used? | Respondent and other person or husband/partner alone or someone else or other or husband/partner has no earnings, respondent alone or respondent and husband/partner |
| | Decision on own healthcare | Who usually makes decisions about healthcare for yourself? | Respondent and other person or husband/partner alone or someone else or other or husband/partner has no earnings, respondent alone or respondent and husband/partner |
| | Decision on large household purchases | Who usually makes decisions about major household purchases? | Respondent and other person or husband/partner alone or someone else or other or husband/partner has no earnings, respondent alone or respondent and husband/partner |
| | Decision on family visits | Who usually makes decisions about visits to your family or relatives? | Respondent and other person or husband/partner alone or someone else or other or husband/partner has no earnings, respondent alone or respondent and husband/partner |
| **Attitudes toward wife beating** | Goes out without telling husband | In your opinion, is a husband justified in hitting or beating his wife if she goes out without telling him? | Yes, No |
| | Neglects children | In your opinion, is a husband justified in hitting or beating his wife if she neglects the children? (v744b) | Yes, No |
| | Refuses sex | In your opinion, is a husband justified in hitting or beating his wife if she refuses to have sex with him? (v744d) | Yes, No |

DHS, Demographic and Health Surveys.

weights and clustering variables. The correlations with the GII were −0.633 ($p = 0.068$) for total empowerment, −0.711 ($p = 0.032$) for "Resources," −0.642 ($p = 0.062$) for "Decision-making," and −0.343 ($p = 0.366$) for "Attitudes toward wife beating" (**Fig B in S5 Appendix**). The strongest, statistically significant correlation with "Resources" was expected, given that women's labor force participation is the only common indicator between the GII and our empowerment score (part of the "Resources" dimension).

## Outcome indicators

We considered 3 sets of indicators: child outcomes (child development and growth), early learning (access to resources and provision of stimulation activities), and nutrition (child dietary diversity). Child development was assessed using the ECDI. ECDI comprises 10 items with response options "yes," "no," and "don't know" administered to the child's mother/caregiver. It assesses 4 development domains: literacy–numeracy (3 items), learning/cognition (2 items), physical development (2 items), and socioemotional development (3 items). Items reflect developmental benchmarks that children are expected to achieve if they are developmentally on track, i.e., they are developing like most of their peers. ECDI is constructed by first scoring individual items as 1 if the child can perform the benchmark, and 0 otherwise. Then, binary indicators are constructed for whether children are developmentally on track in each 1 of the 4 domains. Finally, the ECDI score is calculated as the proportion of children developmentally on track in at least 3 of the 4 domains. Further details on the development, validation, computation, and utilization of the ECDI are available elsewhere [36]. In the present analyses, children were considered offtrack if they failed more than 1 item in each domain [36]. In the appendices, we also present the proportion of children with suboptimal developmental in all

domains. Child growth was assessed using height-for-age Z-score (HAZ), calculated based on the World Health Organization (WHO) Child Growth Standards, and stunting (HAZ < −2 SD) [45].

Early learning opportunities were assessed using 5 indicators for the materials and activities available to the child. One indicator (range 0 to 4) counted the number of materials and resources for child play and learning available in the household: (1) household has at least 1 child book; (2) child plays with homemade toys; (3) child plays with store-bought toys; and (4) child plays with household objects as toys. Four indicators assessed stimulation activities provided by the mother and father in the past 3 days (based on maternal report). These activities or parent–child interactions serve as a common proxy for exposure to early learning opportunities at home [46]. The 6 stimulation activities were as follows: (1) reading books or looking at picture books; (2) telling stories; (3) naming, counting, or drawing with the child; (4) singing songs; (5) taking the child outside the home/yard/enclosure; and (6) playing with the child. We calculated the number of stimulation activities (range 0 to 6) provided separately by each parent and the proportion of parents who provided ≥4 stimulation activities [47]. Finally, child nutrition was assessed using an indicator for dietary diversity score (DDS) and minimum dietary diversity (MDD). Since no validated indicators exist to assess dietary diversity among children 36 to 59 months of age, we used WHO infant and young child feeding indicators for DDS and MDD [48]. Specifically, we constructed DDS (range 0 to 7) by summing the number of food groups consumed by the child in the past 24 hours (based on maternal report) and defined MDD as DDS ≥4. Both indicators have been shown to serve as adequate proxies for micronutrient intake in children 24 to 59 months of age in Burkina Faso [49].

## Statistical analysis

Descriptive statistics were adjusted for the complex survey design using DHS sampling weights. To assess the association between women's empowerment and child development, growth, early learning, and nutrition outcomes, we estimated all models using 2 alternative exposure definitions: (1) one variable for empowerment; and (2) three variables for each individual dimension. Since the associations of interest may be nonlinear, we divided the continuous factor scores derived from the final form-invariant measurement model into country-specific quintile categories. As a secondary analysis, we also estimated the models using the continuous factor scores. Biserial correlations between individual dimensions and total empowerment scores and quintile categories and each outcome are presented in **Tables A and B in S5 Appendix**. For continuous and count outcomes, we fit a generalized linear model and calculated unadjusted and adjusted mean differences (MDs) and 95% confidence intervals (CIs). For binary outcomes, we fit a log-Poisson model and calculated unadjusted and adjusted relative risks (RRs) and their 95% CIs [50]. Given the observational nature of the study, we present adjusted estimates as the primary results and unadjusted estimates in the appendices. Adjusted estimates controlled for the following a priori selected potential confounders: household wealth, rurality, and size; household head's age and sex; woman's education, age, and age at first cohabitation; and child age and sex. We also controlled for country and survey year. Although some measures of women's empowerment include woman's education and age at first cohabitation as indicators [24,51], we excluded them from the empowerment score and treated them as covariates for 2 main reasons. First, it is unclear if women's education and age at first cohabitation represent resources for empowerment, achievement of the empowerment process, or both [52]. Second, women's education is often considered an indicator of women's status and is relatively fixed in adulthood [53]. Missing data on any of the confounders were imputed using mean imputation (N = 6 observations with missing data on household head's

age). Missing data on any of the exposure indicators were handled through the use of a full weight matrix by the CFA model estimator. The proportions of missing data on child development, growth, and early learning outcomes were 4.3%, 4.9%, and 3.2%, respectively, below the 5% threshold typically recommended for complete case analysis [54]. Child diet was collected for a random subsample of children and was therefore missing for 47% of children by design. Thus, only outcome variables had missing values, and none of the exposure or confounder variables used in the models had missing values. Therefore, we used complete case analysis since in this case multiple imputation does not provide any addition information, yields similar estimates if the same predictors of missingness are used, and may in fact introduce uncertainty and increase standard errors [54–56]. All models accounted for clustering and representativeness using the country-specific cluster variables and sampling weights. Lastly, we assessed whether the associations between women's empowerment and child development, growth, early learning, and nutrition outcomes differed across household wealth (defined as a binary variable where 1 = highest 3 quintiles and 0 = lowest 2 quintiles) and woman's education (defined as a binary variable where 1 = any education and 0 = no education). The statistical significance of the interaction was assessed using a Wald test. Associations and interactions were considered statistically significant at $p < 0.05$. All analyses were performed in Stata 16 [57].

## Results

### Sample characteristics

Overall, households were large, primarily living in rural areas, and few were headed by women (**Table 2**). Women were 31 years on average, nearly half had no education, and only one-third had completed primary education. Women were more empowered (mean scores were higher) with respect to the "Resources" and "Decision-making" dimensions than "Attitudes toward wife beating." Nearly one-quarter of children had suboptimal cognitive development, about one-third had suboptimal socioemotional development, 90% had suboptimal literacy–numeracy development, but only 9% had suboptimal physical development. Child growth and dietary diversity were suboptimal too, with over one-third of children stunted and only 14.5% meeting MDD. The number of learning resources and parental stimulation activities was also low.

### Association between women's empowerment and child development and growth

Women's empowerment was weakly associated with child cognitive development with the magnitude of the association similar across quintile categories (**Table 3**). This association was primarily driven by the "Decision-making" dimension with children of women in higher quintile categories being less likely to have suboptimal cognitive development compared to children of women in the lowest quintile category (**Fig 2** and **Table A in S1 Table**). These associations were relatively similar across "Decision-making" quintile categories. In addition, children of women in the third "Attitudes toward wife beating" quintile category were less likely to have suboptimal cognitive development compared to children in the lowest quintile category. With respect to socioemotional development, women's total empowerment was not associated with socioemotional development. However, children of women in the second and fourth "Resources" quintile categories, relative to the first, and children of women in the highest "Attitudes toward wife beating" quintile category, relative to the lowest, were less likely to have suboptimal socioemotional development (**Fig 2** and **Table A in S1 Table**). We found no evidence that women's empowerment and its dimensions are associated with child literacy–numeracy development. Further, we found no evidence of consistent associations between

**Table 2. Characteristics of the women and children included in the analysis sample.**

| | Mean ± SD or Percent |
|---|---|
| *Household characteristics* | |
| N | 21,434 |
| Size | 8.51 ± 5.66 |
| Number of children <5 years | 2.48 ± 1.55 |
| Female-headed | 14.26 |
| Age of household head (years) | 41.71 ± 13.02 |
| Lives in a rural area | 70.88 |
| Poorest wealth quintile | 23.07 |
| *Women's characteristics* | |
| N | 21,434 |
| Age (years) | 30.86 ± 6.62 |
| Highest level of education | |
| No education | 53.00 |
| Primary | 31.27 |
| Secondary | 14.02 |
| Higher | 1.71 |
| Age at first cohabitation (years) | 18.28 ± 4.19 |
| *Empowerment characteristics* | |
| N | 21,434 |
| Resources dimension score | −0.59 ± 1.41 |
| Decision-making dimension score | −0.01 ± 0.67 |
| Attitudes toward wife beating dimension score | −1.68 ± 1.87 |
| Total empowerment score | −2.28 ± 2.92 |
| *Child characteristics* | |
| N | 21,434 |
| Male | 51.08 |
| Age (months) | 47.45 ± 6.96 |
| Development | |
| N | 20,019 |
| Cognitive development offtrack | 22.44 |
| N | 19,688 |
| Socioemotional development offtrack | 30.35 |
| N | 19,335 |
| Literacy–numeracy development offtrack | 89.66 |
| N | 20,082 |
| Physical development offtrack | 9.18 |
| N | 19,255 |
| Overall development offtrack | 13.18 |
| Growth | |
| N | 20,390 |
| HAZ | −1.58 ± 1.4 |
| Stunted (HAZ < −2) | 35.85 |
| Early learning opportunities | |
| N | 21,276 |
| Number of learning resources (0–4) | 1.39 ± 1.04 |
| N | 20,745 |
| Number of maternal stimulation activities (0–6) | 1.5 ± 1.66 |

(*Continued*)

**Table 2.** (Continued)

| | Mean ± SD or Percent |
|---|---|
| ≥4 maternal stimulation activities | 14.06 |
| Number of paternal stimulation activities (0–6) | 0.53 ± 1.07 |
| ≥4 paternal stimulation activities | 3.21 |
| Diet | |
| N | 11,279 |
| DDS (0–7) | 1.68 ± 1.67 |
| MDD (DDS ≥4) | 14.5 |

DDS, dietary diversity score; HAZ, height-for-age Z-score; MDD, minimum dietary diversity.

women's empowerment and child physical development. Children of women in the fourth empowerment quintile category, relative to the first, children of women in the second "Resources" quintile category, relative to the first, and children of women in the second and fourth "Decision-making" quintile categories, relative to the first, were less likely to have sub-optimal physical development (**Fig 2** and **Table A in S1 Table**). Unadjusted estimates and estimates using the continuous scores are shown in **Tables A and B in S1 Table**, respectively.

Children of women in the highest empowerment quintile category, relative to the lowest, had significantly higher HAZ: MD 0.09 (95% CI 0.02, 0.16). We found no evidence that individual dimensions were associated with HAZ or that women's empowerment and its dimensions were associated with child stunting (**Table 3** and **Fig 3** and Table C in **S1 Table**). Unadjusted estimates and estimates using the continuous scores are shown in **Tables C and D in S1 Table**, respectively.

## Association between women's empowerment and early learning opportunities

Women's empowerment was positively associated with early learning opportunities (**Table 4**). Children of women in the highest empowerment quintile category, relative to the lowest, had

**Table 3. Associations between quintile categories of women's total empowerment and child development and growth outcomes[a].**

| | Cognitive development offtrack (N = 20,019) RR (95% CI) | Socioemotional development offtrack (N = 19,688) RR (95% CI) | Literacy–numeracy development offtrack (N = 19,335) RR (95% CI) | Physical development offtrack (N = 20,082) RR (95% CI) | HAZ (N = 20,390) MD (95% CI) | Stunting (HAZ <−2) (N = 20,390) RR (95% CI) |
|---|---|---|---|---|---|---|
| Empowerment Q1 (lowest) | Ref. | Ref. | Ref. | Ref. | Ref. | Ref. |
| Empowerment Q2 | 0.91 (0.82, 1.00) | 1.00 (0.92, 1.09) | 1.00 (0.98, 1.01) | 1.06 (0.88, 1.27) | −0.03 (−0.10, 0.04) | 1.00 (0.94, 1.07) |
| Empowerment Q3 | 0.88 (0.80, 0.98) | 0.93 (0.86, 1.01) | 0.99 (0.98, 1.01) | 0.91 (0.75, 1.10) | 0.01 (−0.06, 0.09) | 0.98 (0.91, 1.04) |
| Empowerment Q4 | 0.89 (0.81, 0.98) | 0.97 (0.89, 1.05) | 0.99 (0.97, 1.00) | 0.82 (0.67, 0.99) | 0.05 (−0.02, 0.13) | 0.97 (0.90, 1.03) |
| Empowerment Q5 (highest) | 0.89 (0.80, 0.99) | 0.93 (0.85, 1.02) | 0.98 (0.96, 1.00) | 0.83 (0.68, 1.02) | 0.09 (0.02, 0.16) | 0.93 (0.87, 1.00) |

[a]All estimates accounted for clustering and representativeness using the country-specific cluster variables and sampling weights and controlled for household wealth, rurality, and size; household head's age and sex; maternal education, age, and age at first cohabitation; child age and sex; and country and survey year.

CI, confidence interval; HAZ, height-for-age Z-score; MD, mean difference; Q, quintile category; Ref, reference; RR, relative risk.

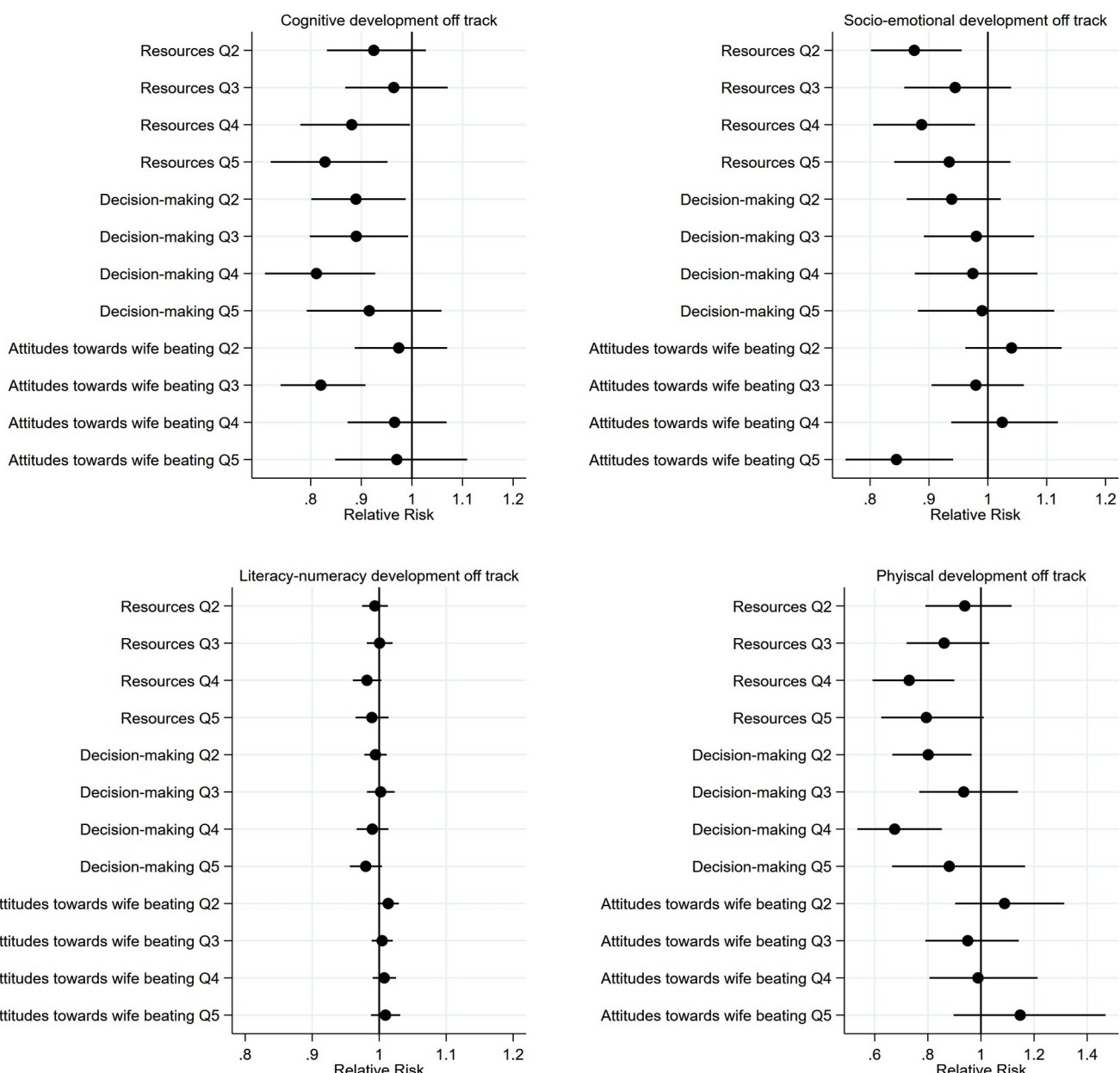

**Fig 2. Associations between quintile categories of women's empowerment dimensions and child development outcomes.** All estimates accounted for clustering and representativeness using country-specific cluster variables and sampling weights and controlled for household wealth, rurality, and size; household head's age and sex; maternal education, age, and age at first cohabitation; child age and sex; and country and survey year. Reference quintile category is Q1, lowest. Q, quintile category.

access to 0.07 additional learning resources (equivalent to 5% additional learning resources) and received 0.16 additional stimulation activities from their mothers (or 12% additional activities), on average. Higher "Resources" and "Decision-making" empowerment was associated with 0.07 to 0.11 additional learning resources (or 5% to 9% more resources) (**Fig 4 and Table A in S2 Table**). Likewise, children of women in higher "Decision-making" quintile categories, relative to the lowest, received 0.18 to 0.34 additional stimulation activities from their mothers (or 15% to 29% more activities). In addition, women in higher "Attitudes toward wife

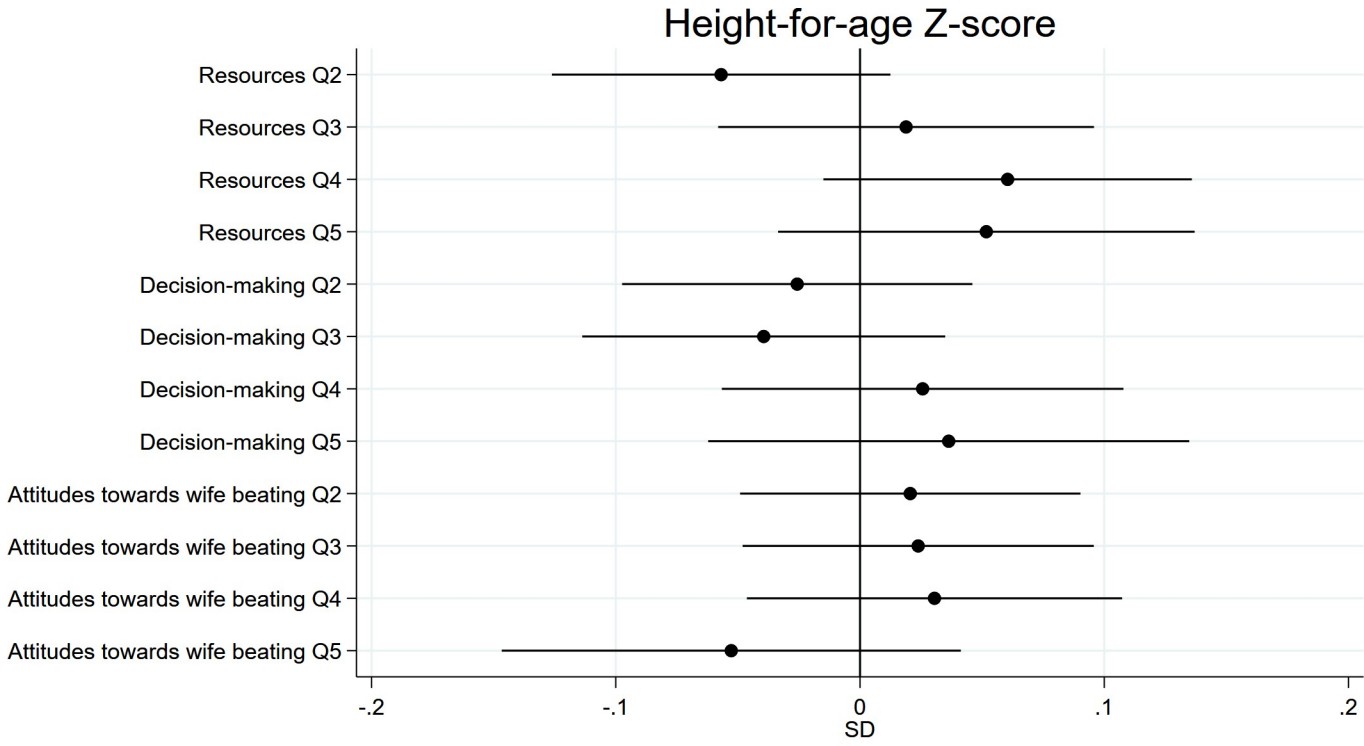

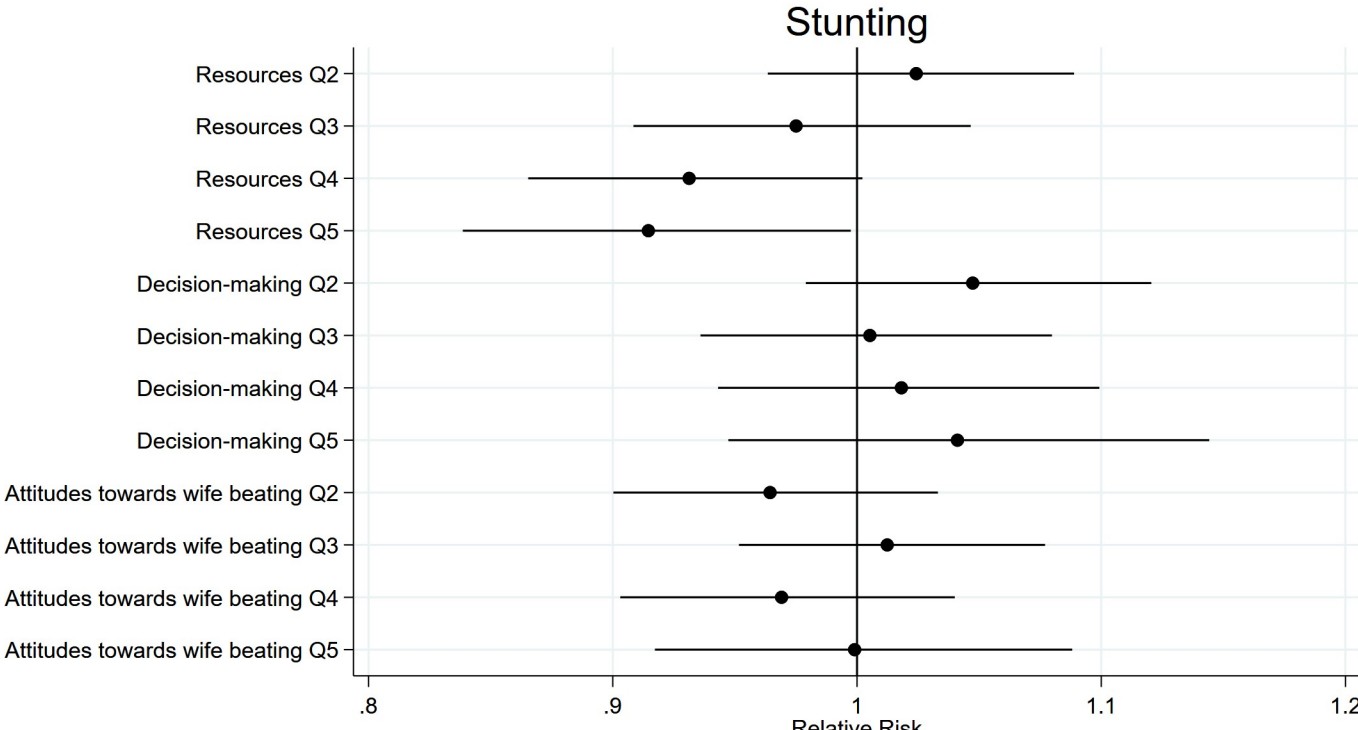

**Fig 3. Associations between quintile categories of women's empowerment dimensions and child growth outcomes.** All estimates accounted for clustering and representativeness using country-specific cluster variables and sampling weights and controlled for household wealth, rurality, and size; household head's age and sex; maternal education, age, and age at first cohabitation; child age and sex; and country and survey year. Reference quintile category is Q1, lowest. Q, quintile category.

**Table 4. Associations between quintile categories of women's total empowerment and early learning and nutrition outcomes[a].**

| | Number of learning resources (0–4) (N = 21,276) MD (95% CI) | Number of maternal stimulation activities (0–6) (N = 20,745) MD (95% CI) | ≥4 maternal stimulation activities (N = 20,745) RR (95% CI) | Number of paternal stimulation activities (0–6) (N = 20,745) MD (95% CI) | ≥4 paternal stimulation activities (N = 20,745) RR (95% CI) | DDS (0–7) (N = 11,279) MD (95% CI) | MDD (DDS ≥4) (N = 11,279) RR (95% CI) |
|---|---|---|---|---|---|---|---|
| Empowerment Q1 (lowest) | Ref. | Ref. | Ref. | Ref. | Ref. | Ref. | Ref. |
| Empowerment Q2 | 0.03 (−0.03, 0.08) | 0.07 (−0.02, 0.16) | 0.98 (0.85, 1.12) | 0.08 (0.03, 0.13) | 1.20 (0.87, 1.67) | 0.06 (−0.06, 0.17) | 1.07 (0.90, 1.28) |
| Empowerment Q3 | 0.02 (−0.03, 0.08) | 0.04 (−0.05, 0.14) | 0.90 (0.78, 1.05) | 0.09 (0.03, 0.14) | 1.14 (0.84, 1.55) | 0.07 (−0.05, 0.18) | 1.02 (0.86, 1.20) |
| Empowerment Q4 | 0.04 (−0.01, 0.10) | 0.09 (0.00, 0.18) | 0.99 (0.86, 1.14) | 0.11 (0.05, 0.16) | 1.22 (0.90, 1.65) | 0.08 (−0.03, 0.20) | 1.01 (0.85, 1.20) |
| Empowerment Q5 (highest) | 0.07 (0.01, 0.13) | 0.16 (0.06, 0.25) | 1.06 (0.91, 1.22) | 0.23 (0.17, 0.29) | 1.79 (1.34, 2.38) | 0.17 (0.06, 0.29) | 1.07 (0.91, 1.27) |

[a]All estimates accounted for clustering and representativeness using the country-specific cluster variables and sampling weights and controlled for household wealth, rurality, and size; household head's age and sex; maternal education, age, and age at first cohabitation; child age and sex; and country and survey year.

CI, confidence interval; DDS, dietary diversity score; MD, mean difference; MDD, minimum dietary diversity; Q, quintile category; Ref, reference; RR, relative risk.

beating" quintile categories were 18% to 26% less likely to provide ≥4 stimulation activities compared to women in the lowest quintile category.

With respect to paternal stimulation activities, we found that more empowered women had more engaged partners. Children of women in higher empowerment quintile categories received 0.08 to 0.23 additional stimulation activities from their fathers, equivalent to 20% to 56% more activities (Table 4). This association was primarily explained by the "Resources" and "Attitudes toward wife beating" dimensions with partners of women in higher quintile categories providing 0.09 to 0.17 (21% to 40%) and 0.05 to 0.13 (13% to 33%) additional stimulation activities, respectively (Fig 5 and Table A in S2 Table). Further, partners whose wives were in the highest empowerment quintile category were 79% more likely to provide ≥4 stimulation activities, and those with wives in the highest "Resources" quintile category were 45% more likely. Unadjusted results and results using the continuous empowerment score are shown in Tables A and B in S2 Table, respectively.

## Association between women's empowerment and child nutrition

Children of women in the highest empowerment quintile category had 0.17 point (or 11%) higher DDS relative to those in the lowest quintile category (Table 4), an association primarily explained by the "Resources" dimension (Fig 6 and Table C in S2 Table). Further, children of women in the highest "Attitudes toward wife beating" quintile category were 22% more likely to meet MDD, relative to the lowest quintile category. Unadjusted estimates and secondary analyses using the continuous scores were generally similar (Tables C and D in S2 Table, respectively).

## Alternative decision-making definition

Results were generally robust to using the alternative decision-making definition, where women were considered empowered if they made decisions alone, rather than alone or together with their partner (S3 Table). In these analyses, the associations between women's empowerment and cognitive development only reached statistical significance when comparing the highest to the lowest quintile categories, whereas the associations between women's

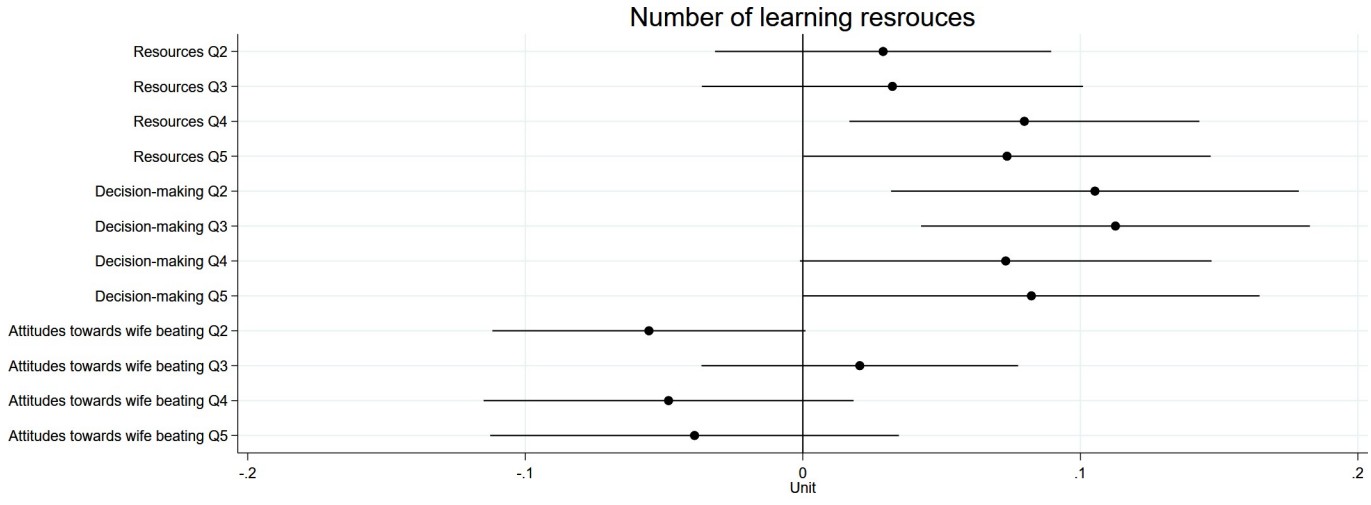

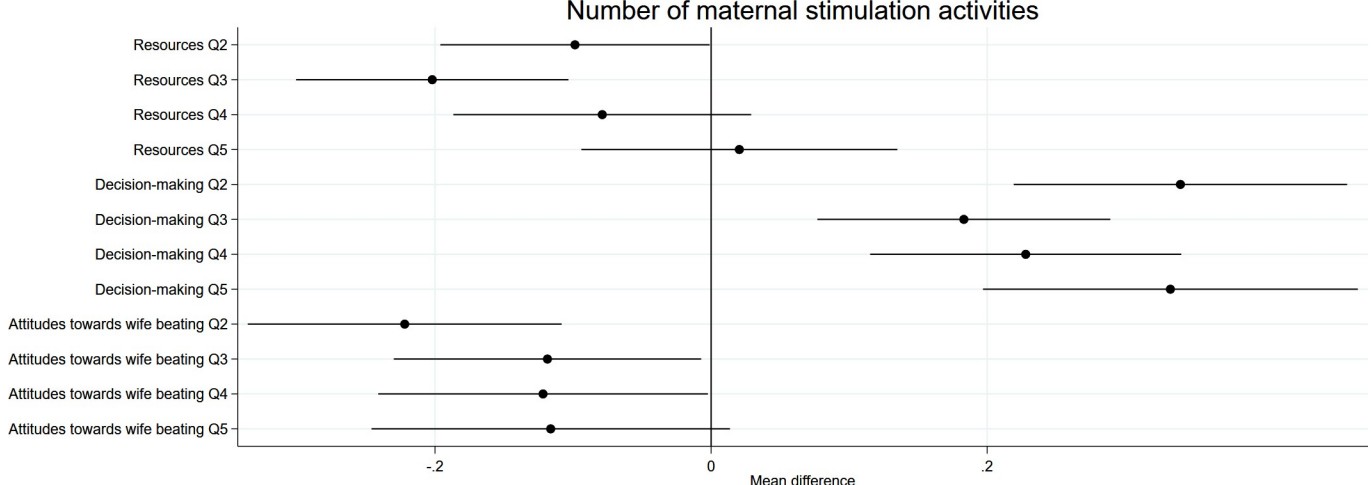

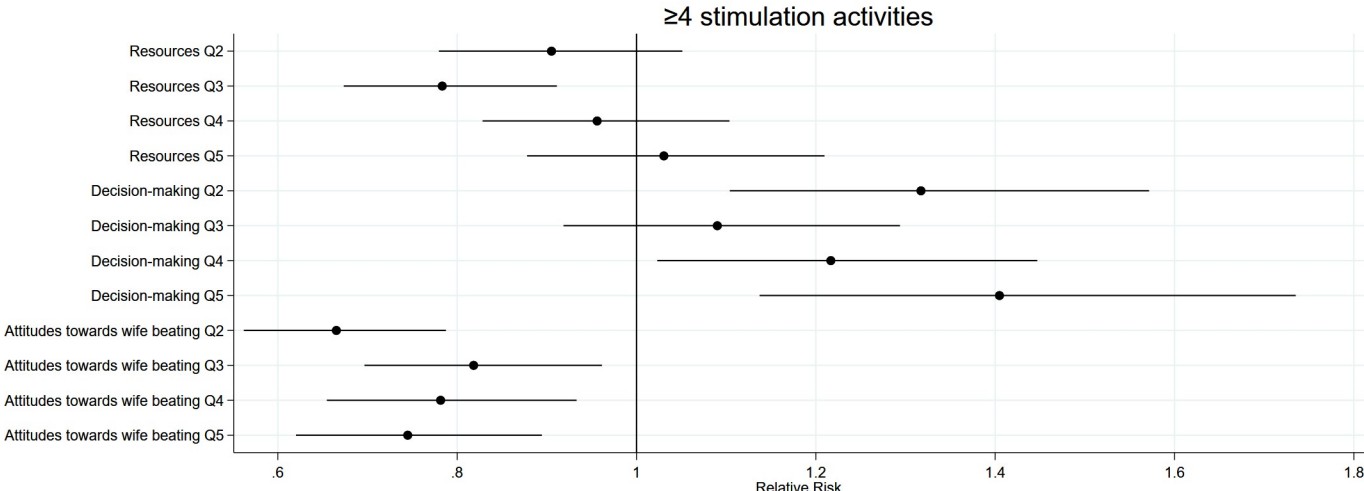

**Fig 4. Associations between quintile categories of women's empowerment dimensions and early learning resources and maternal stimulation outcomes.**
All estimates accounted for clustering and representativeness using country-specific cluster variables and sampling weights and controlled for household wealth, rurality, and size; household head's age and sex; maternal education, age, and age at first cohabitation; child age and sex; and country and survey year. Reference quintile category is Q1, lowest. Q, quintile category.

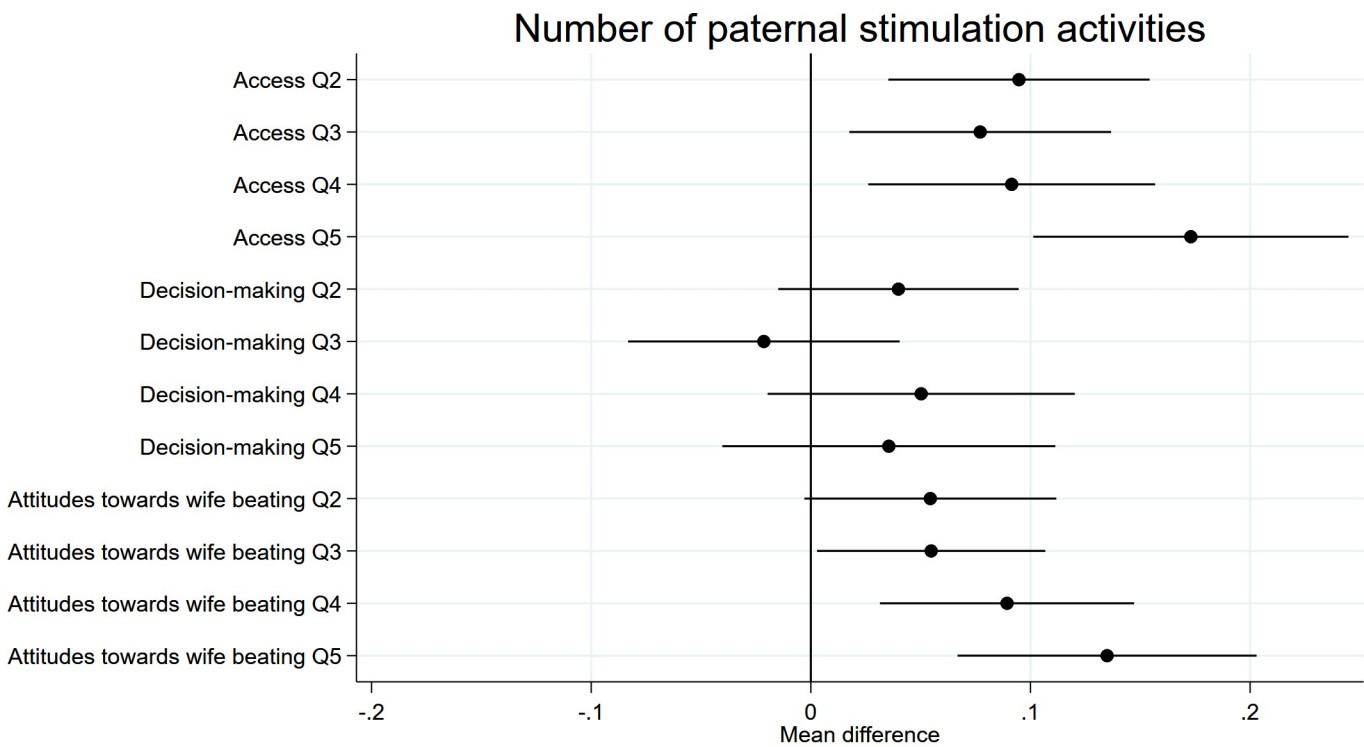

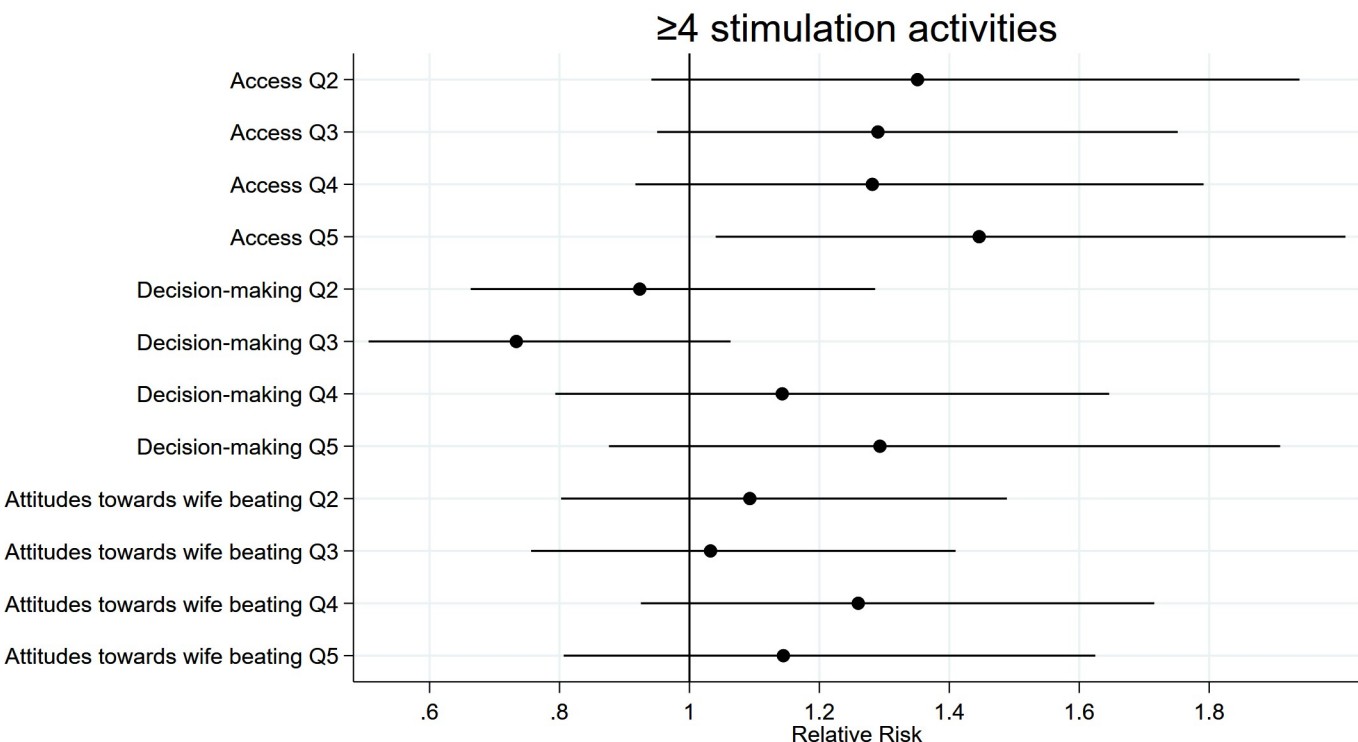

**Fig 5. Associations between quintile categories of women's empowerment dimensions and paternal stimulation outcomes.** All estimates accounted for clustering and representativeness using country-specific cluster variables and sampling weights and controlled for household wealth, rurality, and size; household head's age and sex; maternal education, age, and age at first cohabitation; child age and sex; and country and survey year. Reference quintile category is Q1, lowest. Q, quintile category.

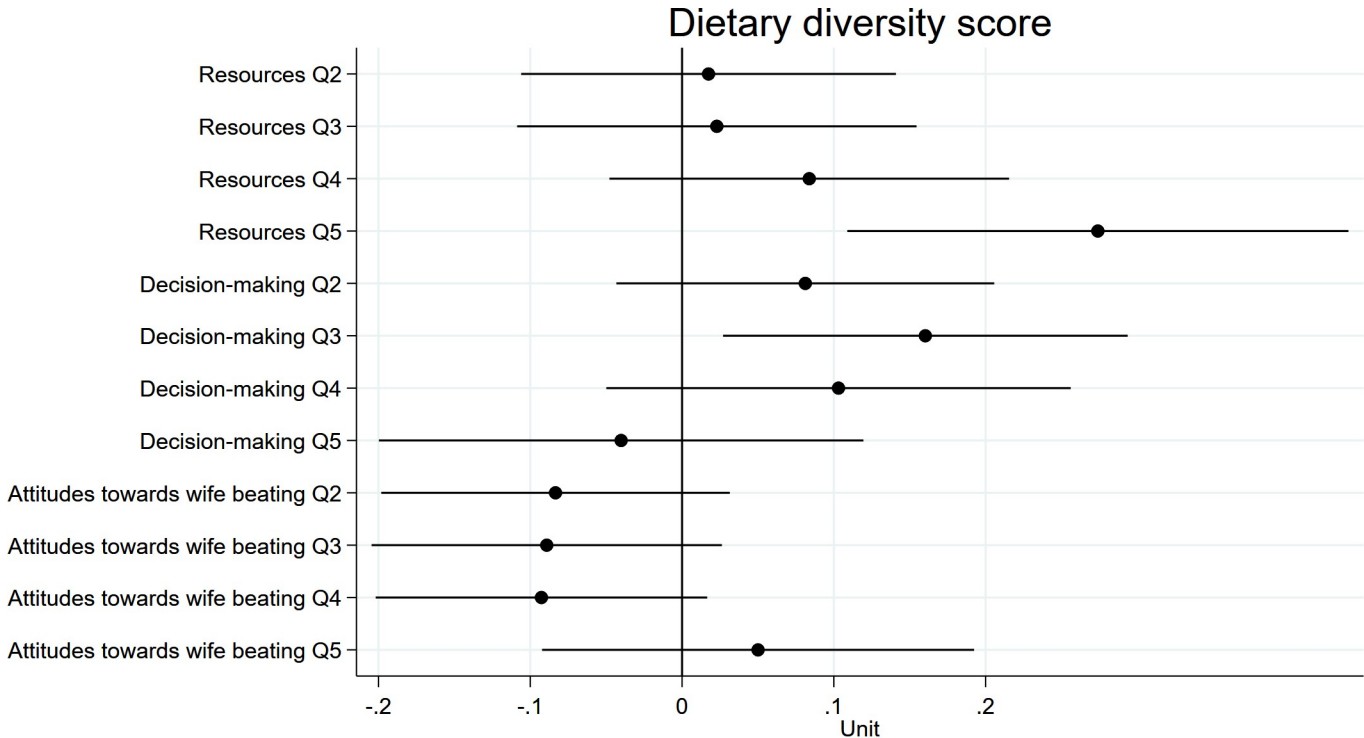

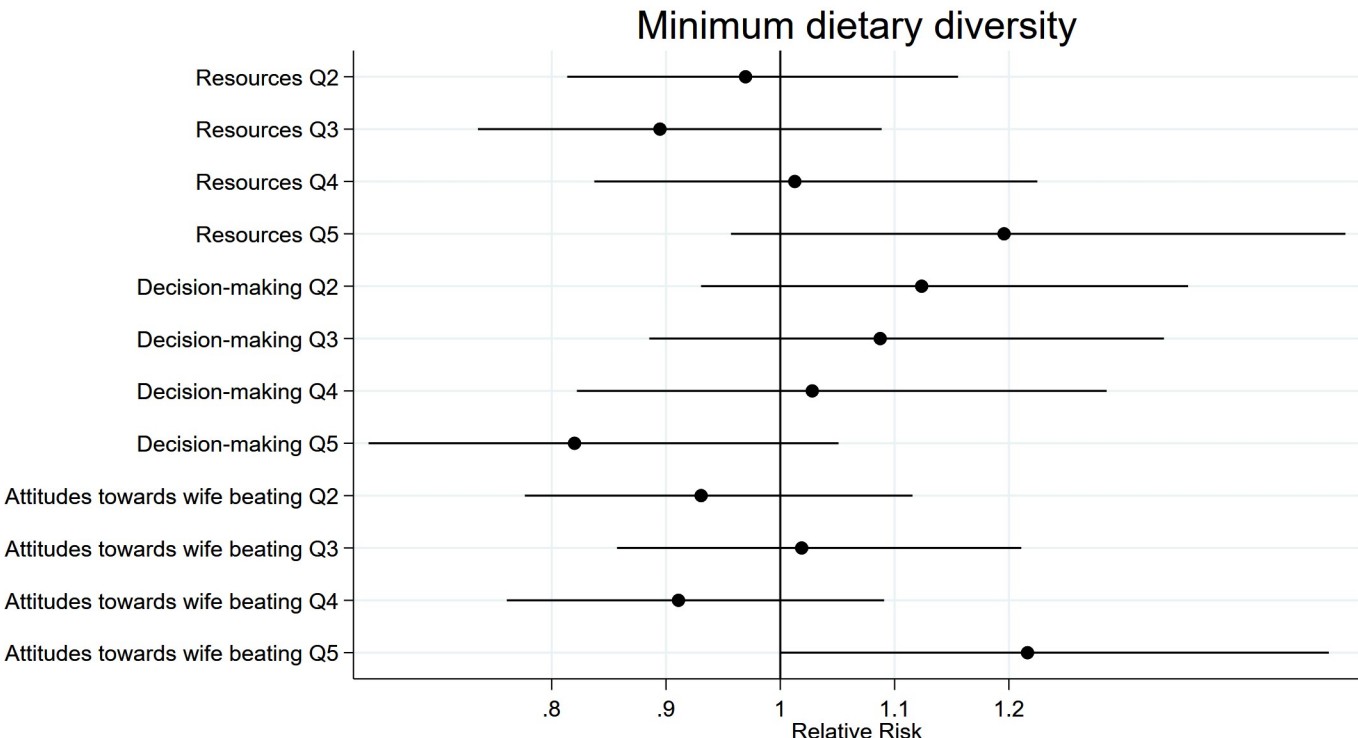

**Fig 6. Associations between quintile categories of women's empowerment dimensions and child nutrition outcomes.** All estimates accounted for clustering and representativeness using country-specific cluster variables and sampling weights and controlled for household wealth, rurality, and size; household head's age and sex; maternal education, age, and age at first cohabitation; child age and sex; and country and survey year. Reference quintile category is Q1, lowest. Q, quintile category.

empowerment and physical development reached statistical significance in the top 2 quintile categories, relative to the lowest (**Table A in S3 Table**). Further, the associations between women's empowerment and number of learning resources and number of maternal stimulation activities did not reach statistical significance (**Table B in S3 Table**). In contrast to the main findings, we found that partners whose wives were in the highest "Decision-making" quintile category provided 0.15 fewer stimulation activities and were 45% less likely to provide ≥4 stimulation activities.

## Heterogeneity

We found that household wealth and women's education modified the association between women's empowerment and child literacy–numeracy development (**S4 Table**). Specifically, the magnitude of the association between women's empowerment and child literacy–numeracy development was larger among wealthy households (RR 0.93 (95% CI 0.91, 0.96)) compared to among poor households (RR 0.99 (95% CI 0.97, 1.01), *p*-value for interaction 0.001) and among women with education (RR 0.92 (95% CI 0.89, 0.95)) compared to women without education (RR 1.00 (95% CI 0.98, 1.02), *p*-value for interaction <0.001). In addition, we found that household wealth and women's education also modified the association between women's empowerment and early learning opportunities. We observed that the magnitude of the association between women's empowerment and number of stimulation activities provided by fathers was larger among wealthy households (MD 0.35 (95% CI 0.27, 0.43)) compared to among poor households (MD 0.13 (95% CI 0.05, 0.20), *p*-value for interaction <0.001). Similarly, the magnitude of the association between women's empowerment and number of paternal stimulation activities was stronger among women with education (MD 0.37 (95% CI 0.27, 0.46)) compared to those without education (MD 0.15 (95% CI 0.09, 0.22), *p*-value for interaction <0.001). Moreover, the association between women's empowerment and number of learning resources was stronger among wealthy households (MD 0.04 (95% CI −0.04, 0.12)) compared to among poor households (MD 0.15 (95% 0.07, 0.24), *p*-value for interaction 0.047) and among women with education (MD 0.17 (95% CI 0.10, 0.25)) compared to women without education (MD 0.05 (95% CI −0.04, 0.13), *p*-value for interaction 0.019). The association between women's empowerment and maternal stimulation activities was also stronger among women with education (MD 0.43 (95% CI 0.30, 0.56)) compared to women without education (MD 0.04 (95% CI −0.10, 0.18), *p*-value for interaction <0.001). Finally, with respect to child cognitive, socioemotional, and physical development, and child growth, and nutrition, we found no evidence of heterogeneity by household wealth or woman's education.

## Discussion

In this study, we found limited evidence suggesting that women's empowerment may be positively associated with child cognitive development, growth, and nutrition outcomes among children 36 to 59 months of age in SSA, and more consistent evidence that women's empowerment and all its dimensions were predictive of learning resources and parental stimulation. However, with respect to child growth and nutrition, evidence was limited with associations significant only among women in the highest empowerment quintile category, relative to the lowest. We found no evidence that women's empowerment or its dimensions were associated with child literacy–numeracy development. With respect to socioemotional and physical development, we only observed associations with specific empowerment dimensions. Different dimensions of women's empowerment were associated with different outcomes: Higher access to and control over resources was predictive of better socioemotional development, whereas higher decision-making power was predictive of better cognitive and physical

development. Overall, most associations were small with relatively wide CIs, indicating that moderate to very small associations were possible. Surprisingly, associations between women's empowerment and child development were not monotonic, i.e., the magnitude of the associations was similar across quintile categories of women's empowerment and its dimensions. In contrast, associations between women's empowerment and parental stimulation generally increased in higher empowerment quintile categories.

To our knowledge, this is only the second study to provide evidence on the association between women's empowerment and child development. Another recent study, which pooled DHS and MICS data from 26 African countries, showed that different aspects of women's empowerment were positively associated with different domains of child development in SSA [20]. Similar to Ewerling and colleagues, we found that women's decision-making empowerment predicted better cognitive development. However, in contrast to this study, we found no evidence that women's empowerment or its dimensions was associated with child literacy–numeracy development, and limited evidence that women's attitudes toward wife beating were associated with socioemotional development. Differences in the operationalization of women's empowerment dimensions, the statistical methods employed, and the sample of countries (in addition to including more SSA countries, Ewerling and colleagues also included 2 North African countries) may help explain the differences in findings between the 2 studies. Of note is that Ewerling and colleagues impute at least some exposure information for nearly 40% of their sample. Although cautiously conducted, this imputation could have biased the association between women's empowerment and child development.

The positive associations between women's empowerment and its dimensions and child cognitive, socioemotional, and physical development we found are plausible, given the observed positive associations between women's empowerment and early learning. Our findings support the hypothesis that women with greater access to and control over resources and greater decision-making power allocate more resources toward their children [11,17]. With respect to socioemotional and physical development, the associations with women's empowerment and its dimensions were limited. Given the positive associations between women's empowerment and early learning, it is possible that ECDI may not adequately capture socioemotional development (with items focused on behavioral challenges) and that nondifferential outcome misclassification would bias the associations with women's empowerment toward the null. In addition, prior studies have shown that the physical and socioemotional indicators of the ECDI are not reliable at the individual level [58]. With respect to literacy–numeracy development, scholars have argued that the ECDI indicators are more advanced than comparable development assessment tools [59], which can help explain the null associations with women's empowerment we observed. Overall, the magnitudes of the associations were relatively small likely due to the long pathways through which women's empowerment affects child development. Nevertheless, our findings contribute to a nascent literature on the relationship between women's empowerment and child development.

A major contribution of our study is establishing an association between women's empowerment and early learning. We demonstrated that women's empowerment was associated with better access to early learning resources, supporting the hypothesis that women's empowerment is crucial in allocating more resources toward children [11,17]. Further, we showed that women's empowerment was associated with more paternal stimulation activities: Partners whose wives were in higher empowerment quintile categories engaged in up to 56% more stimulation activities compared to partners whose wives were in the lowest empowerment quintile category. One potential explanation, compatible with existing literature, is that more empowered women may face time trade-offs due to increased formal or informal labor market participation or improved mobility in their free time [17], thus compromising time allocated

to their children [23]. However, in the absence of time use data, we are unable to empirically confirm this hypothesis. Alternatively, our findings may suggest a shift in traditional caregiving norms. Although more empowered women may remain responsible for domestic work (e.g., preparing meals, cleaning), their partners become responsible for early learning activities (e.g., reading, playing), and women defer to them to provide these activities to children. More empowered women receiving support with childcare are then better able to care for themselves and their families, having sufficient energy, time, and money for domestic work [60].

Surprisingly, women with higher gender empowerment provided fewer stimulation activities, whereas their partners provided more. These results may suggest a change toward more equitable or shared parenting, where more empowered women have more engaged partners. Indeed, in West Africa, empowered women are those who manage the family in tandem with their partners [61]. In South Africa, more empowered women who enter the workforce have more engaged partners who spend more time with their children [62]. However, this relationship may be bidirectional, such that husbands are more engaged to help empower their partners: They treat them as equal, do not submit to social norms that promote gender inequity, and do not leave caregiving to women alone. In parts of Africa, men engage in more childcare to help their wives work [63]. Support of their wives' empowerment may also create more amicable or stable father–mother relationships, which, in turn, can increase paternal engagement [64] and improve fathering [65]. However, caution is warranted in interpreting these results since maternal and paternal stimulation activities were based on maternal self-report and are thus subject to reporting bias. Future research should collect data on stimulation activities directly from men and develop better indicators of men's engagement in childcare, which are currently lacking [11]. More work is also needed to understand and parse out the causal mechanisms behind this relationship between women's empowerment and paternal stimulation.

With respect to child growth, we observed a positive association with women's empowerment in line with existing literature [14,33,66]. In contrast to prior studies where decision-making power predicted better growth outcomes [14,33], total empowerment scores were associated in our study. Multiple differences between prior studies and ours can account for this difference, including different empowerment measures, different child age range, and different study countries. Scholars have previously hypothesized that the association between women's empowerment and child growth is age specific [13], supporting these differential findings. Further, our findings that access to and control over resources and attitudes toward wife beating were not associated with child growth are also in line with existing literature [13]. Importantly, the wide CIs of the associations between women's empowerment and growth outcomes indicated that null/small to large, clinically meaningful associations are plausible. Lastly, of note is that only the highest empowerment quintile category was associated with better child growth. This could be due to the low empowerment levels in our sample, but we also cannot rule out that this association was significant by chance.

In terms of women's empowerment and child nutrition, prior studies have established that women's empowerment is positively associated with child dietary diversity [25,26,30,32,67]. We expand this literature focused on children less than 2 years of age [26,30,32] and preschoolers less than 5 years of age [25,67] by demonstrating that greater women's empowerment and access to and control over resources are associated with improved dietary diversity among children 36 to 59 months of age. Nonetheless, these findings should be interpreted with caution. Similar to our findings on child growth, the association between women's empowerment and child dietary diversity was only significant in the highest empowerment quintile category. Further, the associations between women's decision-making power and child dietary diversity showed no consistent pattern. Lastly, the child dietary diversity indicators we used were validated for use among children 24 to 59 months of age only in Burkina Faso [49], and further

validation in other SSA contexts is needed. Future studies can expand our work by using more comprehensive dietary assessment tools that capture the quantity, quality, and nutrient content of foods consumed by children.

All these findings were based on a decision-making definition treating women as empowered if they made decisions alone or together with their partner. Our results changed little using the alternative decision-making definition treating women as empowered if they made decisions alone. These findings suggest that joint decision-making likely represents cooperation, not disguised male decision-making. Importantly, higher decision-making power predicted fewer paternal stimulation activities in these secondary analyses, highlighting the importance of involving men in the decision-making process in order to increase their engagement in childcare. However, this association was only negative and significant in the highest decision-making quintile category, which could be due to chance.

Lastly, we found that the magnitude of associations between women's empowerment and early learning outcomes was stronger among wealthy households, compared to poor households, and among women with education, compared to women without education. In low-income African communities, this may be due to wealthier households having met immediate health and nutritional priorities [68]. Thus, in wealthier households, women are better able to exercise their empowerment, which is not limited by the household's lack of wealth or resources, and shift time and resources toward early learning activities for their children. These findings are in line with other studies assessing the relationship between women's empowerment and child health and nutrition, which have shown that a certain level of resources or a wealth threshold may be necessary for women to act on their empowerment [66,69]. Additionally, these results lend support to our findings that the magnitude of the association between women's empowerment and child literacy–numeracy development was stronger among wealthy households and among women with education. Together, these findings indicate that poor and uneducated women could potentially benefit more from provision of stimulation inputs such as books, toys, and other manipulatives, or from provision of direct financial and economic resources through cash transfers for example.

Our study has several strengths, including the use of nationally representative data, a conceptual framework grounded in both the women's empowerment and child development literatures, and the use of a measurement invariant empowerment indicator. Despite these strengths, our results should be interpreted with 4 main caveats. First, several exposure and outcome measurement issues should be noted. With respect to women's empowerment, similar to prior studies, we were limited to the indicators collected by the DHS [14,24]. We therefore lack data on direct indicators of all dimensions of women's empowerment relevant to child development such as social resources. Likewise, only indicators on attitudes toward wife beating and no other gender norms are collected. Women could have high empowerment with respect to domestic violence but no empowerment with respect to other gender norms such as domestic work. Further, decision-making indicators were only collected from married women. Women who are single, widowed, or otherwise not married may experience empowerment differently, which would limit the generalizability of our results to these groups of women. In addition, similar to prior studies [24,70], we dichotomized most of the indicators used in our measurement model, which may lead to loss of information [71,72] and implausible models. This latter concern is addressed by our use of a strong conceptual framework, resulting in a form-invariant measurement model with acceptable psychometric properties, despite our use of binary indicators. With respect to child outcomes, ECDI is a crude measure of child development, which does not capture all child development domains, e.g., motor and language development. Future studies should use more comprehensive tools to thoroughly assess the association between women's empowerment and child development. The early

learning outcomes we used are not without limitations either. The parental stimulation indicators only captured 6 activities. Thus, we might not be capturing stimulation activities specific to the SSA context. Although our findings showed that more empowered women had more engaged partners and were themselves less engaged in these 6 stimulation activities, it is possible they provided other stimulation activities not covered by the instrument. Moreover, the indicators we used did not capture the frequency or quality of parental stimulation activities. More research is warranted to more adequately assess the association between women's empowerment and parental stimulation using more comprehensive and objective parental stimulation assessment tools such as the HOME Inventory. Future studies should also test the association between women's empowerment and other caregiver practices and routines to better understand if, and how, women's empowerment can help build and enhance the caregiver environment.

Second, we only have a concurrent measure of women's empowerment, which may introduce time discrepancies in the associations between women's empowerment and child outcomes [11,73]. It is theoretically and empirically unclear whether these associations are cumulative over time or lagged or how early the process of women's empowerment needs to start to influence child outcomes. It is possible that cumulative women's empowerment throughout early life is more important for child outcomes at 36 to 59 months than current women's empowerment. In contrast, early learning and nutrition were assessed over a recent recall period (3 days and 24 hours, respectively). Thus, time discrepancies are less likely, though they cannot be fully dismissed. Longitudinal research is needed to assess how changes over time in women's empowerment influence child, early learning, and nutrition outcomes [13]. Specifically, longitudinal mediation models should explicitly test how the empowerment pathways work individually and together to affect these outcomes and in what temporal order outcomes are affected.

Third, most scholars agree that women's empowerment is context specific [74]. Although we used a measurement invariant empowerment indicator, empowerment may still work in context-specific ways. DHS indicators were designed to measure universal aspects of women's empowerment that are similar across different populations and settings [35]. Thus, it is possible that our measure captured only these similarities in women's empowerment and that context-specific aspects exist that are not captured by the DHS indicators or the measure we derived from them. Indeed, our results showed that the "access to healthcare" dimension may be context specific as this factor did not emerge in 2 of the countries in our sample. Although excluding this dimension may limit the scope of our empowerment score, it may increase its generalizability because it only reflects universal aspects of women's empowerment. Nevertheless, we were limited to 9 SSA countries as we excluded MICS data, which do not collect information on household decision-making or child diet. Thus, we were unable to assess if our findings extend to other SSA countries. Future studies should collect more empowerment indicators and continue to test for measurement invariance across contexts. Common measurement, validation, and analysis approaches of women's empowerment in diverse contexts are needed. Multisite mixed methods or qualitative studies can also help elucidate the universal and context-specific aspects of women's empowerment. More work is needed to assess whether context-specific aspects of women's empowerment differentially affect child, early learning, and nutrition outcomes.

Fourth, the cross-sectional nature of the data does not allow us to establish causality. Our results are subject to reverse causality such that improved child, early learning, and nutrition outcomes may empower women. Future studies should be carefully designed to assess the causal relationships between women's empowerment and these outcomes. Intervention research is needed to test whether improving women's empowerment can also improve child,

early learning, and nutrition outcomes. More measurement research is essential to develop adequate empowerment measures for intervention research, i.e., measures that are sensitive enough to capture changes in response to interventions, as well as measures that can capture the process of empowerment rather than just the state of empowerment.

Despite these limitations, we found suggestive evidence that women's empowerment was positively associated with child development, growth, early learning, and nutrition, though different dimensions were associated with different outcomes. Although much remains to be explored about these associations, our findings have 2 important implications for multigenerational nurturing care interventions to promote child development and growth. First, multigenerational interventions should be designed to promote women's empowerment not only as a potential pathway to improve child, early learning, and nutrition outcomes, but also as an intrinsic benefit rooted in the Sustainable Development Goals. Interventions aiming to improve overall empowerment (i.e., targeting all empowerment dimensions), which was positively associated with the highest number of outcomes in our study, could be more effective than interventions aiming to improve individual empowerment dimensions, which were generally associated with fewer outcomes. When interventions aim to improve specific outcomes, then a heavier focus on specific dimensions may be required, while still continuing to improve overall empowerment [13]. For example, our results indicated that interventions to improve socioemotional development should focus more heavily on the "Resources" and "Attitudes toward wife beating" dimensions but still maintain a general empowerment lens and not overlook other aspects of women's empowerment, which can help improve socioemotional development indirectly by improving early learning and nutrition outcomes.

Second, intervention curricula should be designed to engage women's partners both as a caregiver and as an empowerment champion. Adapting health and education services to include both male and female caregivers can help increase male involvement in childcare [62]. Increasing shared caregiving can help build caregiver capacity and family support [75]. By simultaneously delineating an empowerment champion role, interventions can engage men to provide childcare not only to benefit their children, but also to support their wives. Male engagement should be respectful, supportive, and promotive of women's autonomy, choices, and decision-making [76]. Our results using the 2 different definitions of decision-making showed that positive associations between women's empowerment and paternal stimulation were observed only when men were involved in decision-making. Thus, interventions aiming to increase parental engagement and to promote women's empowerment will likely be more successfully if they also specifically target men. However, to avoid potential negative consequences, women's needs and preferences toward male involvement should also be considered in designing interventions [62].

## Conclusions

In conclusion, we found evidence suggesting that women's empowerment was positively associated with child cognitive development in SSA, but not child socioemotional, literacy–numeracy, or physical development. The association between women's empowerment and child cognitive development is plausible, given the positive associations with the underlying nurturing care outcomes, i.e., early learning and nutrition. Household wealth and woman's education modified the association between women's empowerment and child literacy–numeracy development and early learning. Different empowerment dimensions predicted different outcomes. Future research should focus on improving measurement, exploring longitudinal associations, and establishing causality. Interventions to promote child development and growth should engage men to increase women's empowerment and gender equity with respect to child care and support.

## Supporting information

**S1 STROBE Checklist. STROBE Checklist.**
(DOCX)

**S1 Text. Definition and measurement of women's empowerment.**
(DOCX)

**S1 Appendix. Methods to derive women's empowerment factor scores. Table A.** Coding of indicators from Demographic and Health Survey (DHS) variables used to describe the dimensions and subdimensions of women's empowerment. **Table B.** Indicators of women's empowerment excluded from the exploratory factor analysis and reasons for exclusion. **Table C.** Proportion of women endorsing each indicator of women's empowerment.
(DOCX)

**S2 Appendix. Exploratory factor analysis results. Fig A.** Scree plot from exploratory factor analysis conducted on one random split-half sample. **Table A.** Exploratory factor analysis of women's empowerment dimensions on one random split-half sample.
(DOCX)

**S3 Appendix. Confirmatory factor analysis results. Table A.** Confirmatory factor analysis of women's empowerment dimensions on one random split-half sample by country.
(DOCX)

**S4 Appendix. Multigroup confirmatory factor analysis results. Table A.** Estimated factor correlation matrices for the latent variables from the form-invariant confirmatory factor analysis model by country.
(DOCX)

**S5 Appendix. Empowerment score diagnostics. Fig A.** Distributions of the individual dimension factor scores and the total empowerment score overlayed with the standard normal distribution. **Fig B.** Correlation between individual dimension and total empowerment scores and the Gender Inequality Index. **Table A.** Biserial correlations between the continuous total empowerment and individual dimensions scores and all considered outcomes. **Table B.** Biserial correlations between the total empowerment and individual dimensions quintile categories and all considered outcomes.
(DOCX)

**S1 Table. Association of women's empowerment and child development and growth. Table A.** Associations between quintile categories of women's total empowerment and empowerment dimensions and child development. **Table B.** Associations between women's total empowerment and empowerment dimensions and child development. **Table C.** Associations between quintile categories of women's total empowerment and empowerment dimensions and child growth. **Table D.** Associations between women's total empowerment and empowerment dimensions and child growth.
(DOCX)

**S2 Table. Association of women's empowerment and early learning and nutrition outcomes. Table A.** Associations between quintile categories s of women's total empowerment and empowerment dimensions and early learning outcomes. **Table B.** Associations between women's total empowerment and empowerment dimensions and early learning outcomes. **Table C.** Associations between quintile categories of women's total empowerment and empowerment dimensions and child diet. **Table D.** Associations between women's total

empowerment and empowerment dimensions and child diet.
(DOCX)

**S3 Table. Results using the alternative definition of decision-making indicators. Table A.**
Associations between women's total empowerment and empowerment dimensions (continuous and quintile categories) and child outcomes, using the alternative definition of decision-making indicators. **Table B.** Associations between women's total empowerment and empowerment dimensions (continuous and quintile categories) and care outcomes, using the alternative definition of decision-making indicators.
(DOCX)

**S4 Table. Effect heterogeneity of the association between women's total empowerment and child and care outcomes by household wealth and maternal education, comparing women the highest quintile category to women in the lowest quintile category.**
(DOCX)

## Acknowledgments

We would like to thank all the participants in the studies and the DHS Program teams that implement, conduct, and complete the DHS surveys and make the data available.

## Author Contributions

**Conceptualization:** Lilia Bliznashka, Ifeyinwa E. Udo, Christopher R. Sudfeld, Wafaie W. Fawzi, Aisha K. Yousafzai.

**Data curation:** Lilia Bliznashka.

**Formal analysis:** Lilia Bliznashka.

**Methodology:** Lilia Bliznashka.

**Supervision:** Christopher R. Sudfeld, Wafaie W. Fawzi, Aisha K. Yousafzai.

**Visualization:** Lilia Bliznashka.

**Writing – original draft:** Lilia Bliznashka.

**Writing – review & editing:** Lilia Bliznashka, Ifeyinwa E. Udo, Christopher R. Sudfeld, Wafaie W. Fawzi, Aisha K. Yousafzai.

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
