## [Editor Report · Decision Letter 0]

7 Dec 2020

Dear Dr Bliznashka, 

Thank you for submitting your manuscript entitled "Women’s empowerment is associated with improved child development, growth, and nurturing care practices in sub-Saharan Africa" for consideration by PLOS Medicine.

Your manuscript has now been evaluated by the PLOS Medicine editorial staff, as well as by an academic editor with relevant expertise, and I am writing to let you know that we would like to send your submission out for external peer review.

Kind regards,

Artur A. Arikainen,

Associate Editor

PLOS Medicine

---

## [Decision Letter · Decision Letter 1]

16 Jan 2021

Dear Dr. Bliznashka,

Thank you very much for submitting your manuscript "Women’s empowerment is associated with improved child development, growth, and nurturing care practices in sub-Saharan Africa" (PMEDICINE-D-20-05677R1) for consideration at PLOS Medicine. 

Your paper was evaluated by a senior editor and discussed among all the editors here. It was also discussed with an academic editor with relevant expertise, and sent to three independent reviewers, including a statistical reviewer (r#1 - see pdf review). The reviews are appended at the bottom of this email and any accompanying reviewer attachments can be seen via the link below:

[LINK]

In light of these reviews, I am afraid that we will not be able to accept the manuscript for publication in the journal in its current form, but we would like to consider a revised version that addresses the reviewers' and editors' comments. Obviously we cannot make any decision about publication until we have seen the revised manuscript and your response, and we plan to seek re-review by one or more of the reviewers. 

We expect to receive your revised manuscript by Feb 08 2021 11:59PM. Please email us (plosmedicine@plos.org) if you have any questions or concerns.

We look forward to receiving your revised manuscript. 

Sincerely,

Emma Veitch, PhD

PLOS Medicine

On behalf of Clare Stone, PhD, Acting Chief Editor, 

PLOS Medicine

plosmedicine.org

*Please revise your title according to PLOS Medicine's style - ideally this should include a designation of the study design in the title (normally we recommend this is in the subtitle, after a colon) - here the study is based on DHS surveys, which are cross-sectional, correct? So one option might be:

"Association between women’s empowerment and child development, growth, and nurturing care practices in sub-Saharan Africa: analysis of cross-sectional survey data".

*In the last sentence of the Abstract Methods and Findings section, please include a note about any key limitation(s) of the study's methodology; in this case this might include the possibility for reverse causality or unmeasured confounding.

*Did your study have a prospective protocol or analysis plan? Please state this (either way) early in the Methods section.

*As an analysis of cross-sectional survey data (the DHS surveys), we'd recommend that the authors use the STROBE guideline to support reporting of the study. In this case please include the completed STROBE checklist as Supporting Information. Please add the following statement, or similar, to the Methods: "This study is reported as per the Strengthening the Reporting of Observational Studies in Epidemiology (STROBE) guideline (S1 Checklist)."

Comments from the reviewers:

Reviewer #1: See attachment

Michael Dewey

Reviewer #2: 

Overall, this study is excellent study and the manuscript well-written. The methods and interpretation were thoughtful and meticulously presented.

1. Line 152. Use "resources" for brevity rather than "access" for two reasons. One, both access and control are important. Two, then this language would better articulate with Kabeer and with the concept of resources for care in the UNICEF extended model for care (reference 19).

2. Line 261. "with an identity like"? Authors meant "link"? Why not just say "we fit a general linear model"?

3. Lines 377-378. Prior research with MICS data has shown that the socio-emotional indicators in the ECDI are not reliable at the individual level. That knowledge should be brought into the discussion here. See the ECDI measures section of: Frongillo EA, Kulkarni S, Basnet S, de Castro F. Family care behaviors and early childhood development in low- and middle-income countries. Journal of Child and Family Studies 26:3036-3044, 2017.

Reviewer #3: 

PMEDICINE-D-20-05677R1

Women's empowerment vs child development and growth

1. This is an important analysis that contributes to the growing literature on women's empowerment using survey data. 

2. The literature review on which the article is based seems outdated, however. The authors cite a "large body of literature" on women's empowerment and child health and nutrition (l.81), but the latest article is from 2016. A lot has happened since then. 

3. Next, the authors claim that no study has looked at the association between empowerment and child development. It's true for the limited list of articles cited. But they missed "The impact of women's empowerment on their children's early development in 26 African countries." published in the Journal of Global Health, https://doi.org/10.7189/jogh.10.020406. 

4. In short, the introduction misses several more recent publications on women's empowerment and several other aspects of child health and survival using survey data, giving the reader the false impression that the topic is understudied. 

5. Figure 1 shows pathways through which empowerment would affect child development (and growth? not included in the caption). The first thing that calls my attention is that many of the arrows are bidirectional. I find it hard to interpret, and mostly to implement in any type of analysis. I also do not recognize the most commonly described dimensions of empowerment in this model. 

6. It is not clear from the methods section why only DHS surveys were used in the analysis. The decision is surprising given that MICS was the survey family that started applying the ECDI module, and using MICS surveys would largely increase the number of countries represented in the SSA region. 

7. Regarding the empowerment indicator used, despite the criticisms raised by the authors against other attempts, they provide details on what was developed in a series of appendices. The indicator was not published and subject to direct screening by peers. The review of the indicator would deserve a full evaluation. In my opinion the validation presented relies strongly in psychometric approaches, where some ideal statistics are sought. But we do not see any evidence that the resulting scores have any meaning as measures of empowerment in the form of some external validation with other existing indicators. 

8. The ECDI has been heavily criticized in the literature, and alternative ways to assessing child development have been proposed, besides the original proposal. Here, the authors decided to drop two of the original domains and keep the social-emotional and cognitive domains. The reasons presented, that one set of items is too difficult and the other is too easy do not convince me. Checking the proportion of children failing these dimensions confirm what the authors say, but in including these items, the indicator as a whole has more information to discriminate children that are not on track. In my opinion, dropping information from what is available only makes the indicator weaker. The decision basically relies on one publication that is very critical of the indicator. If the authors agree with that paper, one wonders if the indicators should have been used here at all. 

9. The empowerment scores were divided into quintiles. It would be interesting to have an idea of the score distribution. FA scores based on dichotomous variables can pile up in a few values and make the use of quintiles very hard to interpret. 

10. It would be good to cite a source justifying the use of a Poisson model for binary outcomes, even if this practice is quite common currently. 

11. The list of confounders include, in my opinion, several variable that actually constitute aspects of empowerment, like education and age at first marriage. 

12. I find the presentation of results hard to follow. Especially figures 2 and 3, with very large Cis and no clear trend, do not help understand whatever the data may be suggesting. It is not specified in the figures that the development indicators are actually children not on track. A less careful examination of the figures suggest that empowerment is detrimental to development. 

13. With all due respect to the authors, I cannot agree with the first statement in the discussion. The results show many more null associations than positive associations, and where there is some association, they are quite weak and small. This is more likely due to the indicators used than the lack of effect of women's empowerment. 

14. Again, in line 368, the assertion needs to be corrected.

[LINK]

---

## [Decision Letter · Decision Letter 2]

19 Aug 2021

Dear Dr. Bliznashka,

Thank you very much for re-submitting your manuscript "Associations between women’s empowerment and child development, growth, and nurturing care practices in sub-Saharan Africa: a cross-sectional analysis of Demographic and Health Survey data" (PMEDICINE-D-20-05677R2) for consideration at PLOS Medicine. We do apologize for the delay in sending you a response.

I have discussed the paper with our academic editor and it was also seen again by two reviewers. I am pleased to tell you that, provided the remaining editorial and production issues are fully dealt with, we expect to be able to accept the paper for publication in the journal.

[LINK]

We hope to receive your revised manuscript within 2 weeks. Please email us (plosmedicine@plos.org) if you have any questions or concerns.

Please let me know if you have any questions, and we look forward to receiving the revised manuscript.   

Sincerely,

Richard Turner PhD

rturner@plos.org

Requests from Editors:

In the submission form, you mention that restrictions exist on data access, but are these data not freely available from DHS? Please amend the statement either to explain briefly what the restrictions are, or alter it to "Yes - all data freely available" or similar.

Please remove the information on funding, competing interests and data access from the title page. This information will appear in the article metadata in the event of publication, via entries in the submission form. 

Acknowledgements should appear at the end of the main text.

Please list the country names in the abstract.

Please quote aggregate demographic details for study participants in the abstract.

Please correct the typo in reference 70.

Comments from Reviewers:

*** Reviewer #1: 

[See attachment]

Michael Dewey

*** Reviewer #3: 

I congratulate the authors for the thorough response for the issues I raised, and the careful and detailed revision of the paper. I have no more issues, and I believe the paper can be published at this stage.

***

[LINK]

---

## [Editor Report · Decision Letter 3]

25 Aug 2021

Dear Dr Bliznashka, 

On behalf of my colleagues and the Academic Editor, Dr Persson, I am pleased to inform you that we have agreed to publish your manuscript "Associations between women’s empowerment and child development, growth, and nurturing care practices in sub-Saharan Africa: a cross-sectional analysis of Demographic and Health Survey data" (PMEDICINE-D-20-05677R3) in PLOS Medicine.

PRESS

Sincerely, 

Richard Turner, PhD 

rturner@plos.org